# Amine-Functionalized Maghemite Nanoflowers for Efficient Magnetic Removal of Heavy-Metal-Adsorbed Algae

**DOI:** 10.3390/ijms262010010

**Published:** 2025-10-15

**Authors:** Tímea Fóris, Péter Koska, Ágnes Maria Ilosvai, Kitti Gráczer, Ferenc Kristály, Lajos Daróczi, Miklós Nagy, Béla Viskolcz, László Vanyorek

**Affiliations:** 1Institute of Chemistry, University of Miskolc, Miskolc-Egyetemváros, 3515 Miskolc, Hungary; timea.foris1@uni-miskolc.hu (T.F.); peter.zoltan.koska@uni-miskolc.hu (P.K.); maria.agnes.ilosvai@uni-miskolc.hu (Á.M.I.); kitti.graczer@uni-miskolc.hu (K.G.); miklos.nagy@uni-miskolc.hu (M.N.); 2Higher Education and Industrial Cooperation Centre, University of Miskolc, Miskolc-Egyetemváros, 3515 Miskolc, Hungary; 3Institute of Mineralogy and Geology, University of Miskolc, Miskolc-Egyetemváros, 3515 Miskolc, Hungary; ferenc.kristaly@uni-miskolc.hu; 4Department of Solid-State Physics, University of Debrecen, P.O. Box 2, 4010 Debrecen, Hungary; lajos.daroczi@science.unideb.hu

**Keywords:** *Chlorella vulgaris*, amine-functionalized maghemite nanoflowers, cobalt adsorption, magnetic separation

## Abstract

This study aimed to clarify the interactions between cobalt-adsorbed *Chlorella vulgaris* cells and amine-functionalized maghemite nanoparticles, focusing on nanoparticle adsorption to the algal surface and the subsequent magnetic sedimentation of the formed complexes. The combined process of cobalt uptake by algae and secondary binding of magnetic nanoparticles demonstrates a promising and sustainable strategy for heavy metal removal from industrial wastewater. The adsorption capacity of *Chlorella vulgaris* was assessed, achieving 96 ± 2% Co^2+^ removal, followed by magnetic separation using γ-Fe_2_O_3_ nanoparticles. The subsequent magnetic separation of the cobalt-adsorbed biomass achieved efficiencies ranging from 57.43% to 97.64% within a 60 s timeframe, demonstrating a significant enhancement over conventional sedimentation methodologies. Stable nanoparticle–biomass binding was facilitated by electrostatic interactions between protonated amine groups on the surface of amine-functionalized maghemite particles and the negatively charged functional groups of the algal cell wall, complemented by the contribution of hydroxyl and carboxyl groups. The even distribution of amine-functionalized maghemite nanoparticles on algal surfaces was further validated by Transmission Electron Microscopy (TEM) imaging, and the strong magnetic properties of the nanoparticles enabled rapid and efficient separation under an external magnetic field. This study underscores the promise of integrating *Chlorella vulgaris* with amine-functionalized maghemite nanoparticles as a cost-effective, biocompatible, and environmentally sustainable approach for large-scale heavy metal removal from industrial wastewater.

## 1. Introduction

The presence of heavy metals in natural waters and soils, resulting from industrial discharges, poses a significant environmental concern. The battery industry is a significant contributor to this phenomenon, given that the effluents from battery manufacturing and recycling facilities commonly contain substantial quantities of these metals. The cathode materials of lithium-ion batteries employed in the automotive sector (e.g., NMC-Nickel Manganese Cobalt, NCA-Nickel Cobalt Aluminum and LMO-Lithium Manganese Oxide chemistries) incorporate varying ratios of nickel, manganese, and cobalt [1]. While elevated concentrations of heavy metals exert acute toxic effects on living organisms, even trace levels are hazardous due to their capacity for bioaccumulation and subsequent transfer through the food chain [2].

A variety of technologies have been employed for the removal of heavy metals from industrial wastewater, including ion exchange, reverse osmosis, electrodialysis, and ultrafiltration [3]. Despite their efficacy, these methodologies are often accompanied by substantial limitations, including incomplete metal removal, the formation of sludge, elevated reagent and energy utilization, aggregation of metal precipitates, and membrane fouling. Consequently, there has been an increased focus on alternative treatment strategies, with bioremediation emerging as a highly promising option [4]. The increasing interest in bioremediation can be attributed to its potential advantages in terms of cost-efficiency, effectiveness, and environmental sustainability [5]. This approach employs biological systems to reduce pollutant concentrations and toxicity to acceptable levels. The microorganisms, encompassing bacteria, algae, and yeasts, function as the predominant agents in these processes. This is attributable to their ability to adsorb toxic metal ions onto their cell surfaces and accumulate them in substantial amounts within their intracellular spaces [6,7]. In this group, the freshwater microalga *Chlorella vulgaris* has exhibited remarkable potential for the removal of heavy metals [8].

The cell wall of *Chlorella vulgaris* is primarily composed of complex carbohydrates, consisting of a cellulose fibre framework cross-linked with polysaccharides such as hemicellulose and glycoproteins [9]. In addition, it contains various monosaccharides, including glucose, mannose, galactose, xylose, fucose, and arabinose, together with lipids and substantial amounts of glucosamines. Functional groups present within these biopolymers play a critical role in metal ion adsorption. Specifically, the algal cell wall incorporates carboxyl, hydroxyl, amino, ester, sulfhydryl, sulphate, and phosphate groups. Above pH 3, the deprotonation of carboxyl, phosphate, and hydroxyl groups imparts a net negative surface charge, which facilitates electrostatic interactions with positively charged heavy metal cations [10]. Since functional group protonation is pH-dependent, the surface charge of the cell wall varies with pH [11]. Heavy metal binding to the algal surface biopolymers can proceed through multiple mechanisms, including physisorption, ion exchange, chelation, and complexation [12].

While the removal of heavy metal ions from industrial wastewater is critical, the subsequent recovery of algal biomass that has accumulated these metals must also be addressed. Conventional harvesting methods include sedimentation, flotation, coagulation–flocculation, filtration, and repeated sedimentation steps [13]. Among emerging alternatives, magnetic separation offers distinct advantages owing to its rapid operation, cost-effectiveness, and low energy requirements. Algal cells exhibit a strong affinity for magnetisable nanoparticles (MNPs), which readily adsorb onto their cell wall surfaces, thereby allowing efficient recovery under the influence of an external magnetic field [14,15]. Compared with conventional sedimentation, which generally requires more than two hours for effective coagulation and settling, magnetic separation achieves comparable efficiencies within approximately 60 s [16].

Maghemite (γ-Fe_2_O_3_) nanoparticles possess stable and strongly magnetic properties, allowing for their rapid and efficient separation from solutions using an external magnetic field [17]. Their excellent adsorption capacity is due to their highly specific surface area and porosity, making them effective at binding cell walls of microalgae [18]. Compared to other nanoparticles, they are less toxic, making them suitable for both environmental and biomedical applications [19]. Their surfaces can be easily functionalized, enhancing their selectivity and broadening their range of uses. These characteristics make maghemite nanoparticles a promising alternative, particularly in fields like wastewater treatment [20].

The central aim of this research was to elucidate the interactions between cobalt-adsorbed *Chlorella vulgaris* cells and maghemite nanoparticles, with particular emphasis on the adsorption of the nanoparticles onto the algal cell surface and the subsequent sedimentation of these complexes under the influence of an external magnetic field. By combining the initial adsorption of cobalt, which represents a typical heavy metal contaminant originating from battery manufacturing effluents, with the secondary binding of magnetic nanoparticles, this study provides new insights into the development of an integrated and sustainable approach for the removal and recovery of heavy metals from industrial wastewater. The interactions between microalgal cells and magnetic nanoparticles were examined by Scanning Electron Microscopy (SEM), while Fourier Transform Infrared Spectroscopy (FTIR) was employed to identify the functional groups involved. High-Resolution Transmission Electron Microscopy (HRTEM) was used to characterize the particle size and morphology of the magnetic nanoparticles.

## 2. Results and Discussion

### 2.1. Characterization of the Maghemite Nanoflowers

HRTEM measurements were carried out for the study of particle size and morphology. The TEM pictures show the spherical morphology of the amine-functionalized maghemite nanoflowers; at a higher resolution, you can see the characteristic fine structure that gives them the name of nanoflower (Figure 1A,B). The average size of these maghemite nanoflowers is 226 ± 66 nm, but this is only characteristic of the secondary structure; in fact, these aggregate structures are composed of smaller crystallites (Figure 1C).

The special structure is of great importance, given that the self-assembly of numerous crystallites has resulted in the creation of micropores. Energy dispersive X-ray spectroscopy (EDS) analysis was also performed on the maghemite sample (Figure 1D). The EDS found iron and oxygen, the chemical elements that make up maghemite (γ-Fe_2_O_3_) or magnetite (Fe_3_O_4_). In addition to the elements oxygen and iron, carbon has also been identified, which is partly due to the organic molecules which are adsorbed by the maghemite nanoflowers. The detected presence of copper and carbon is attributable to the composition of the sample holder (copper grid with lacey carbon). The positions of the amine-functionalized maghemite elements are clearly visible on the elemental maps (Figure 1E). It was observed that only the maghemite nanoflowers exhibited detectable traces of iron and oxygen, indicating that iron is not present in its elemental state but occurs in the sample as iron oxide, specifically maghemite. Carbon can also be detected in the micropores of the maghemite nanoparticles and on its surface, due to the adsorbed organic molecules (ethylene glycol and monoethanolamine). During the Transmission Electron Microscopic examination, Selected Area Electron Diffraction (SAED) measurement was applied to the study of the crystalline phase in the nanoflowers (Figure 2A,B). Based on the SAED results, the d-spacing values were calculated, which correspond to the values given by the X-ray databases for maghemite (PDF 39-1346).

Following confirmation that the nanoflowers observed in the TEM images were composed of maghemite crystallites, and in order to eliminate the possibility of non-magnetic contaminants, X-ray Diffraction (XRD) measurements were performed. Reflections characteristic of the maghemite and magnetite phases were found on the XRD diffractogram. The Miller indices corresponding to the reflections were identical to those indicated on the SAED image. The XRD pattern of the sample (Figure 3) shows those reflections, which are characteristic of the maghemite (γ-Fe_2_O_3_); these are located at 18.3° (111), 23.8° (210), 26.1° (211), 30.2° (220), 35.6° (311), 37.2 (222), 43.3° (400), 53.7° (422), 57.2° (511), and 62.8° (440) two theta degrees, in agreement with the PDF 39–1346 card. In addition to the 91.8 wt% maghemite, another magnetic iron oxide, namely magnetite (Fe_3_O_4_), is present in the sample in an amount of 8.2 wt%. The reflections characteristic of magnetite at 18.2° (111), 30.1° (220), 35.4° (311), 37.1 (222), 43.1° (400), 53.4° (422), 56.9° (511), and 62.6° (440) can be found in the diffractogram (PDF 19–0629). Other iron oxide phases (hematite) or other salts (e.g., sodium chloride) as residue were not detected; in this sense, the synthesis method can be efficiently used for the preparation of maghemite nanoflowers.

The –NH_2_, –OH, and –COOH functional groups present on the surface of the amine-functionalized maghemite particles play a crucial role in the adsorption processes between the magnetic nanoparticles and algal cells; therefore, FTIR measurements were performed to identify these interactions (Figure 4). Two characteristic bands were identified on the FTIR spectrum of maghemite at 452 cm^−1^ and 587 cm^−1^ wavenumbers, which were assigned to intrinsic stretching vibration modes of the metal–oxygen bonds (νM-O) at the octahedral and tetrahedral sites [21]. The peaks at 1017 cm^−1^ and 1068 cm^−1^ belong to the νC-O and νC-N vibration in the adsorbed monoethanolamine [22,23]. Another absorption band was observed at 1163 cm^−1^, which can be assigned to the out-of-plane deformation (twisting) mode of –CH_2_ groups, while its asymmetric deformation vibrational counterpart (γ_s_CH_2_) appeared at 1458 cm^−1^. The other visible band at 1391 cm^−1^ can belong to the OH bending vibrations in the adsorbed ethylene glycol molecules. The band βNH_2_ vibration mode was found at 1547 cm^−1^ wavenumber; it corresponds to the free amine functional groups in MEA [24]. Strazisar et al. demonstrated that the bending vibrational mode of the amine groups (between 1580 cm^−1^ and 1630 cm^−1^) has been shown to interfere with the bending vibration mode of the adsorbed water molecules at 1641 cm^−1^ and causes spectral distortion [25]. The symmetric and asymmetric stretching vibrations of aliphatic C–H bonds produced peaks at 2857 cm^−1^ and 2930 cm^−1^, which can be attributed to organic molecules such as ethylene glycol and monoethanolamine adsorbed on the surface of the maghemite nanoflowers [26,27]. In addition, the stretching vibration bands of hydroxyl and amine groups overlap, giving rise to a broad absorption region between 3000 and 3750 cm^−1^.

The zeta potential of the amine-functionalized maghemite nanoparticles exhibited a clear pH dependence. Under acidic conditions (pH < 5), the particles carried a net positive charge due to the protonation of surface amine groups (–NH_3_^+^), while at around pH 4.5–5 the zeta potential approached zero, indicating the isoelectric point. At pH 6–6.5, the zeta potential of the amine-functionalized maghemite suspension was slightly negative (−2.4 to −10 mV), primarily due to the deprotonation of surface hydroxyl and carboxyl groups. Nevertheless, locally protonated amine groups remain present and provide positively charged sites that promote binding to negatively charged functional groups on the *Chlorella vulgaris* cell wall (ζ = −20.1 ± 0.5 mV). The primary driving force for binding is therefore electrostatic attraction between these protonated amines on the nanoparticles and the deprotonated groups on the algal surface, further complemented by hydrogen bonding.

The magnetic behaviour of the maghemite sample was characterized at room temperature using Vibrating Sample Magnetometry (VSM). The saturation magnetization (Ms) of the sample was 71.1 emu/g at 10 kOe magnetic field (Figure 5). A comprehensive review of the extant literature reveals a broad spectrum of Ms values ranging from 10 to 90 emu/g. Our maghemite nanoflowers are commensurate with this range [28,29]. Our Ms value is slightly smaller than the reported Ms value of the bulk maghemite, which is 75–92 emu g^−1^ at room temperature [30]. The magnetization curve exhibits a narrow hysteresis loop with low coercivity (Hc = 160 Oe) and low remanent magnetization (Mr = 10.8 emu/g). These relatively small values confirm the ferromagnetic behaviour of the synthesized particles at room temperature. Furthermore, the narrow hysteresis loop indicates that the samples can be easily demagnetized, as clearly demonstrated in Figure 5.

### 2.2. Results of the Cobalt Adsorption Tests Using Chlorella vulgaris

Cobalt elicited only a marginal, non-significant inhibition of growth relative to the control. Following a 24 h period of inoculation, there was an observed increase in the initial OD680 values from 0.5 to 1.67 (Figure 6A). The exponential phase was observed to last for a period of 72 h following inoculation, at which point it transitioned into the plateau phase. The highest OD680 value recorded was 6 at this time-point. The cobalt concentration does not exhibit a significant decrease during the eight-hour period of algal cell growth because the number of cells available for bioaccumulation is insufficient to take up the cobalt ions (Figure 6A,B). Following a 24 h period of inoculation, a substantial increase in the quantity of algae binding significant amounts of cobalt is observed. The initial Co^2+^ concentration in the medium was 41.3 mg/L, which decreased to 1.8 ± 0.6 mg/L after 24 h, corresponding to a removal efficiency of 96 ± 2%.

### 2.3. Results of Magnetic Separation of Cobalt-Adsorbed Chlorella vulgaris

Magnetic separation was accomplished with high efficiency at low algal biomass concentrations, specifically 0.9 g/L (OD_680_ = 0.5) and 1.8 g/L (OD_680_ = 1.0), using 4 mL suspensions (Figure 7A). Under these conditions, harvesting efficiencies of 97.64% and 94.61% were obtained, respectively. In contrast, at higher algal biomass concentrations, the recovery rates within 60 s decreased to 75.25% and 57.43% (Figure 7C,D). Extending the magnetization time did not improve the separation efficiency, indicating that binding between nanoparticles and algal cells rapidly reached equilibrium. These observations suggest that, at low cell densities, amine-functionalized maghemite nanoparticles readily interact with and adhere to the algal cell surfaces, resulting in near-complete recovery. At higher algal concentrations, however, the reduced efficiency is likely attributable to the limited availability of nanoparticles relative to cell surface binding sites. Consequently, it can be anticipated that increasing the nanoparticle concentration would enhance recovery efficiency under these conditions.

The addition of amine-functionalized maghemite nanoparticles to the cobalt-binding algae was followed by a separation process from the purified water sample via magnetic separation. The presence of cobalt was confirmed through a combination of electron microscopic examination, elemental mapping and energy dispersive X-ray spectroscopy (EDS). As illustrated in the high-angle annular dark-field (HAADF) image (Figure 8A), the larger spherical structures correspond to *Chlorella vulgaris* cells, while the amine-functionalized maghemite nanoparticles appear as smaller bright spots with higher contrast attached to the algal surface. The element map shows iron enrichment, which corresponds to the location of the maghemite nanoparticles on the algae cells (Figure 8B). The elemental map also shows the position of cobalt, indicating that cobalt is detectable across the entire surface of the algae cells, which also confirms that *Chlorella vulgaris* is an excellent choice for removing cobalt through bioaccumulation. The presence of cobalt on the surface of the algae was also confirmed by EDS measurements (Figure 8C). Furthermore, maghemite was also well-bound on the surface of the algae, and, therefore, by applying a magnetic field, the algae could be effectively recovered from the purified water sample.

## 3. Materials and Methods

### 3.1. Materials

The maghemite nanoflowers were synthesized from iron (II) chloride tetrahydrate, FeCl_2_ ∙ 4 H_2_O, MW: 198.81 g/mol (VWR Int. Ltd., B-3001 Leuven, Belgium) and iron (III) chloride, anhydrous, FeCl_3_, MW: 162.20 g/mol (VWR Int. Ltd., B-3001 Leuven, Belgium). Ethylene glycol, HOCH_2_CH_2_OH, (VWR Int. Ltd., F-94126 Fontenay-sous-Bois, France) was applied as solvent. For amine functionalization and for coprecipitation of the nanoparticles, monoethanolamine (MEA), NH_2_CH_2_OH (Merck KGaA, D-64271 Darmstadt, Germany) and sodium acetate, CH_3_COONa (ThermoFisher GmbH, D-76870 Kandel, Germany) were used. The culture medium for *C. vulgaris* consisted of the following materials: potassium nitrate, KNO_3_, potassium dihydrogen phosphate, KH_2_PO_4_, magnesium sulphate, MgSO_4_·7 H_2_O, citric acid, iron(II) sulphate, FeSO_4_·7 H_2_O, calcium chloride, CaCl_2_·2 H_2_O, ethylene diamine tetraacetic acid, sodium salt, (NaOOCCH_2_)_2_NCH_2_CH_2_N(CH_2_COONa)_2_·H_2_O, orthophosphoric acid, H_3_BO_3_, zinc sulphate, ZnSO_4_·7 H_2_O, manganese chloride, MnCl_2_·4 H_2_O, sodium molybdate Na_2_MoO_4_·2 H_2_O, and copper sulphate, CuSO_4_·5 H_2_O (Sigma-Aldrich Ltd., Saint Louis, MO 63103, USA)

### 3.2. Synthesis of the Amine-Functionalized Maghemite Nanoflowers

The solvothermal synthesis of the maghemite happened in a Teflon-lined hydrothermal autoclave (with 150 mL volume), at 200 °C, for 12 h. In the first step, iron (II) chloride tetrahydrate (18 mmol) and anhydrous iron (III) chloride (36 mmol) were solved in ethylene glycol (100 mL). In the second step, sodium acetate (90 mmol) was added to the solution of iron precursors and stirred at room temperature until dissolution of the CH_3_COONa, followed by the addition of 60 mL (0.9921 mol) monoethanolamine. The solution was transferred into the autoclave, after which it was heated at 200 °C for 12 h. After, from the cooled dispersion, the solid phase was separated by a magnet from the ethylene glycol phase, and it was washed with distilled water, and finally rinsed with ethanol (96 Vol%). The maghemite nano powder was dried at 90 °C overnight.

### 3.3. Cobalt Adsorption Tests

The *Chlorella vulgaris* strain was obtained from the Advanced Materials and Intelligent Technologies Higher Education and Industrial Cooperation Centre at the University of Miskolc. The *C. vulgaris* strain was maintained in modified “endo” medium [31] containing KNO_3_ 3 g/L, KH_2_PO_4_ 1.2 g/L, MgSO_4_·7H_2_O 1.2 g/L, citric acid 0.2 g/L, FeSO_4_·7H_2_O 0.016 g/L, CaCl_2_·2H_2_O 0.105 g/L, and trace element stock solution 1 mL/L. For trace elements, a stock solution was prepared containing Na_2_EDTA 2.1 g/L, H_3_BO_3_ 2.86 g/L, ZnSO_4_·7H_2_O 0.222 g/L, MnCl_2_·4H_2_O 1.81 g/L, Na_2_MoO_4_·2H_2_O 0.021g/L, and CuSO_4_·5H_2_O 0.07g/L. After the solution of all components of the medium, the pH was adjusted to 6.5 ± 0.2 and the temperature was controlled at 24 °C.

For the cultivation of *C. vulgaris*, a glass tubular air-lift photobioreactor system was constructed. The setup included a Hailea V20 membrane compressor (20 L/min airflow capacity, 15 W power) connected to nine glass vessels, each with a working volume of 500 mL. The airflow rate supplied to each vessel was 4.5 L/min. Illumination was provided by LED lamps with a light intensity of 3000 lumens. The reactors were inoculated with a concentrated *C. vulgaris* culture containing 5 × 10^8^ cells/mL at an inoculation ratio of 1%, resulting in an initial cell density of approximately 5 × 10^7^ cells/mL in each reactor vessel.

To study the cobalt bioaccumulation process with *C. vulgaris* microalga, cobalt stock solution was measured into the culture medium prior to inoculation. Thus, the initial Co^2+^ concentration of the medium reached 41.3 mg/L ± 3.6. After cobalt addition, the medium was inoculated with *C. vulgaris* culture with a 2% inoculation volume ratio. The initial cell density could be characterized by OD680 value 0.5 ± 0.06.

To study the cobalt removal, kinetics samples were taken for Co^2+^ analysis immediately after cobalt addition from the uninoculated medium, then 2, 4, 6, 8, 24, 48, 72, and 96 h after inoculation. The algae that bound cobalt (II) ions were removed from the suspension using a syringe filter (0.2 micron) after sampling.

The concentration of cobalt ions in the supernatant was measured by inductively coupled plasma atomic emission spectrometry (ICP-AES) using a Varian 720-ES simultaneous multielement spectrometer equipped with an axial plasma view. The operating parameters were as follows: RF generator frequency, 40 MHz; sample introduction device, V-groove nebulizer with Sturman–Masters spray chamber; sample uptake rate, 2.1 mL/min; signal integration time, 8 s; and three replicate readings per sample. Calibration was carried out with a series of standard solutions prepared from a 1000 mg/L single-element cobalt stock solution (Certipur, Merck Ltd., Darmstadt, Germany).

Co^2+^ removal efficiency was calculated by the following Equation:$$\mathrm{Removal}\mathrm{efficiency}\%\;=\;\frac{{{[Co}^{2+}]}_{t0}-{\left[{Co}^{2+}\right]}_{t1}}{{{[Co}^{2+}]}_{t0}}\;\times \;100$$

[Co^2+^]_t0_ = initial concentration of Co^2+^ given in mg/L.

[Co^2+^]_t1_ = Co^2+^ concentration at the given time point after cobalt addition of the cell free supernatant.

### 3.4. Magnetic Separation Tests of the Cobalt-Bonded Algae

The time and efficiency of magnetic separation of cobalt-adsorbed algal biomass were evaluated at different algal concentrations. A concentrated *Chlorella vulgaris* culture containing 4.2 mg/L dry biomass was diluted to obtain suspensions of varying concentrations: 3.6 mg/L (OD_680_ = 2.0 ± 0.1), 2.7 mg/L (OD_680_ = 1.5 ± 0.1), 1.8 mg/L (OD_680_ = 1.0 ± 0.1), and 0.9 mg/L (OD_680_ = 0.5 ± 0.1). The concentration of maghemite nanoparticles was kept constant at 0.511 g/L. To each 100 mL algal suspension of different concentrations, cobalt solution was added to achieve a final concentration of 60 mg/L, followed by mixing with air bubbling for 30 min.

Subsequently, the algal biomass was treated with maghemite nanoparticles at a concentration of 0.511 g/L for 10 min using the air bubbling compressor. The pH was adjusted to 6.5 with phosphate buffer and the temperature was 24 °C.

To assess the binding efficiency of nanoparticles to cobalt-loaded algal cells, the sedimentation rate under a magnetic field was measured. For this purpose, 4 mL of algal suspension was transferred into a custom-made cuvette (10 × 10 × 3 mm) equipped with an N45 neodymium magnet fixed at the base. The cuvette was placed in the spectrophotometer holder, and sedimentation was monitored by recording changes in optical density over time. Measurements were performed at 680 nm using a UV1100/UV-1200 double-beam spectrophotometer (Frederiksen).

The harvesting efficiency (HE%) of the process was calculated from the following Equation:$$\mathrm{HE}\%\;=\;\frac{{OD}_{0}-{OD}_{1}}{{OD}_{0}}\;\times \;100$$

OD_0_ is the initial absorbance of microalgae cultivation at a wavelength of 680 nm.

OD_1_ is the absorbance at the same wavelength of the supernatant liquid that separates from the microalgae particle flocs after the application of the magnetic field.

### 3.5. Characterization Techniques

The morphology, fine structural features, and particle size distribution of the maghemite nanoflowers were characterized by High-Resolution Transmission Electron Microscopy (HRTEM, Talos F200X G2, field emission electron gun X-FEG, accelerating voltage: 20–200 kV; Thermo Scientific, Waltham, MA, USA). Particle sizes were determined from the TEM images using the scale bar and pixel ratios with ImageJ (Fiji) win64 software [32]. Crystalline phase identification of individual nanoparticles was performed by Selected Area Electron Diffraction (SAED), recorded with a SmartCam digital search camera (Ceta 16 Mpixel, 4k × 4k CMOS; Thermo Scientific, Waltham, MA, USA) in combination with a high-angle annular dark-field (HAADF) detector. For the electron microscopy examination, the maghemite particles were suspended in dist. water, then dropped onto a copper grid and dried (Ted Pella Inc., 4595 Redding, CA 96003, USA). X-ray diffraction (XRD) measurement and Rietveld analysis were used for the identification of the phase composition of the maghemite sample. The XRD examination by Bruker D8 diffractometer (Cu-Kα source) was carried out in parallel-beam geometry (Göbel mirror) with Vantec detector. Fourier Transform Infrared Spectroscopy (FTIR) was employed to investigate the amine-functionalized maghemite nanoflowers, with the aim of identifying the surface functional groups. The spectroscopic measurement was performed with Bruker Vertex 70 instrument (Bruker Optics GmbH & Co. KG, 76275 Ettlingen, Germany) in transmission mode, in KBr pellet (10 mg maghemite in 200 mg KBr). The electrokinetic potential of the maghemite particles was determined from electrophoretic mobility using laser Doppler electrophoresis with an Anton Paar Litesizer DLS 500 instrument (Anton Paar GmbH, Graz, Austria). The magnetic properties of the sample, including saturation magnetization (Ms), remanent magnetization (Mr), and coercivity (Hc), were analyzed with a self-developed vibrating sample magnetometer (VSM) equipped with a water-cooled Weiss-type electromagnet (University of Debrecen). For the VSM measurements, the sample was pelletized (20 mg), and the magnetization (M) was recorded as a function of the applied magnetic field (H) up to 150,000 A/m at room temperature.

## 4. Conclusions

We have prepared amine-functionalized maghemite nanoparticles that can be effectively used to magnetically separate bioaccumulating algae that have bound toxic heavy metal ions from water. The bioaccumulation study of the algae demonstrates their ability to bind large amounts of cobalt from water samples, reducing the Co^2+^ concentration from 41.3 mg/L to 0.5 ± 0.1 mg/L within 48 h, achieving a 99 ± 0.2% removal efficiency. Analysis of the maghemite nanoflowers with a vibrating sample magnetometer (VSM) revealed that they can be easily separated together with the algae due to their high saturation magnetization (Mr: 71.1 emu/g). Magnetic deposition studies of the algae also confirmed the rapid magnetic separation; the algal biomass attached to maghemite nanoparticles was separated from the nutrient solution with an efficiency of 57.43–97.64% within 60 s. This efficiency was found to be concentration-dependent, i.e., it varied according to the initial biomass concentration. The elemental maps obtained from the electron microscopy studies show that the adsorbed cobalt is homogeneously distributed on the surface of the algae and maghemite nanoparticles were also identified on their surface. Although the measured zeta potential of the amine-functionalized maghemite suspension at pH 6.5 was slightly negative, this does not exclude the presence of locally protonated amine groups. Given that primary amines have pK_a_ values around 9–10, they remain largely protonated near neutral pH, providing positively charged surface sites [33]. Several studies have reported that aminated iron oxide nanoparticles can still exhibit negative overall zeta potentials due to the influence of other surface functionalities (e.g., hydroxyl or carboxyl groups), yet localized protonated amines persist and contribute to electrostatic attraction with negatively charged biomaterials [34,35]. This surface heterogeneity means that zeta potential values represent an average charge state and may not fully reflect local binding environments. Consequently, the strong attachment of amine-functionalized maghemite nanoparticles to the negatively charged *Chlorella vulgaris* cell wall (*ζ* = −20.1 ± 0.4 mV at pH 6.5) can be explained by the presence of these localized positive sites that drive electrostatic interactions, possibly complemented by hydrogen bonding [36]. These combined effects enable stable nanoparticle–biomass binding even when the overall suspension zeta potential is negative.

These results underscore the potential of low-cost, biocompatible, and widely available magnetic nanoparticles in wastewater treatment, particularly for the removal of heavy metals, presenting a promising approach for sustainable environmental remediation. An equally significant aspect for practical application is the regeneration or safe disposal of the heavy-metal-loaded algal biomass. Regeneration can be achieved through the desorption of bound metals using dilute mineral acids, chelating agents such as EDTA, or other suitable eluents [37]. This process allows the algal biomass to be reused in multiple adsorption–desorption cycles, thereby reducing overall treatment costs. In instances where regeneration is not a viable option, safe disposal strategies must be implemented. These may include immobilization of the biomass in stable solid matrices to prevent leaching, or thermal treatment/incineration, where metals can be concentrated and potentially recovered from the resulting ash [38]. Subsequent research will concentrate on methodically evaluating these options to enhance both environmental safety and cost-effectiveness.

## Figures and Tables

**Figure 1 ijms-26-10010-f001:**
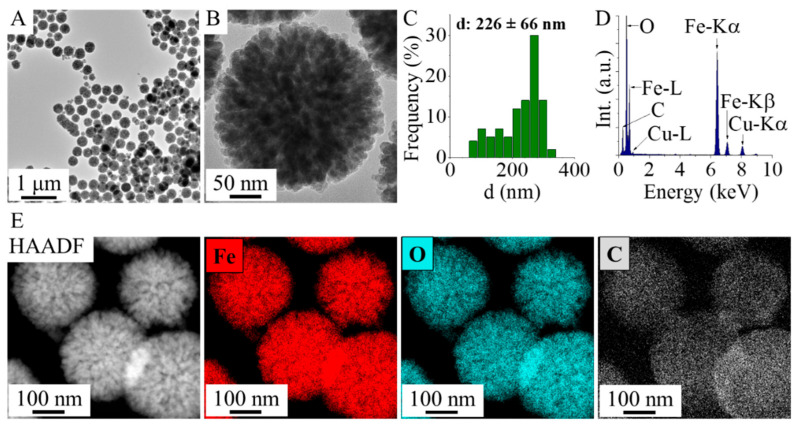
HRTEM pictures (**A**,**B**), size distribution (**C**), EDS spectrum (**D**), and HAADF picture with element maps of the maghemite nanoflowers (**E**).

**Figure 2 ijms-26-10010-f002:**
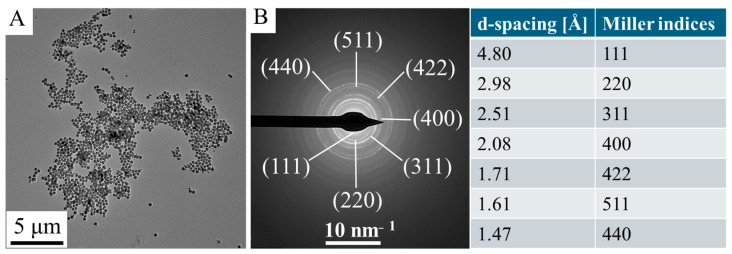
TEM pictures (**A**) SAED picture and the d-spacing values with Miller indices (**B**) of the maghemite nanoflowers.

**Figure 3 ijms-26-10010-f003:**
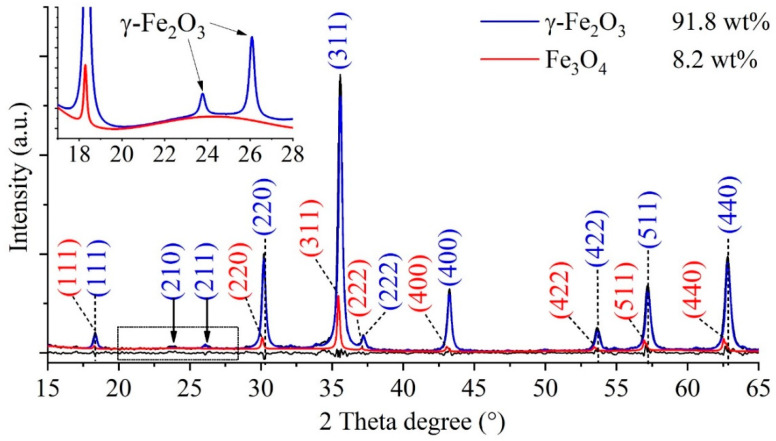
X-ray diffraction of the maghemite.

**Figure 4 ijms-26-10010-f004:**
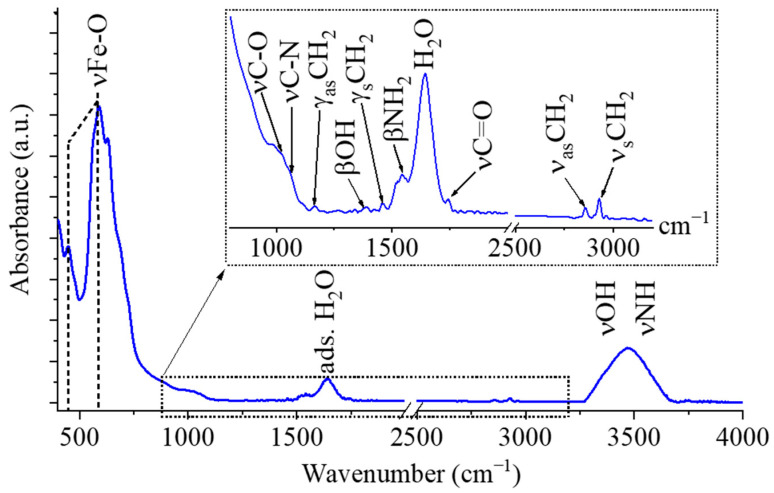
FTIR spectra of the maghemite nanoflowers.

**Figure 5 ijms-26-10010-f005:**
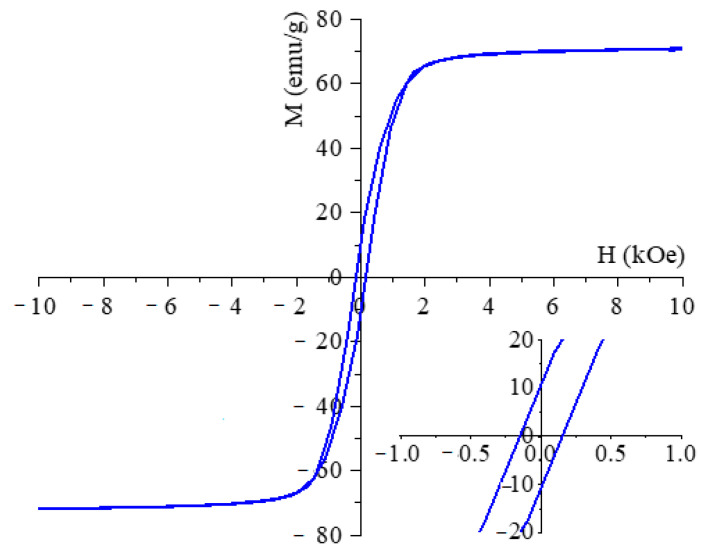
VSM curve of the maghemite nanoflowers.

**Figure 6 ijms-26-10010-f006:**
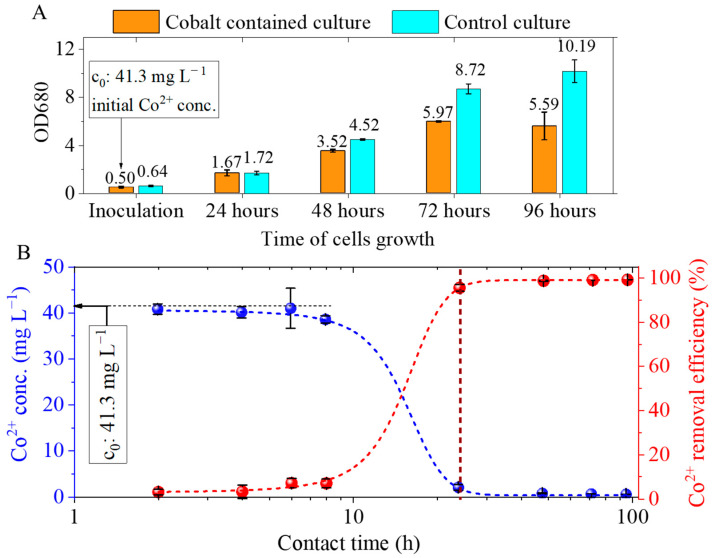
Cobalt adsorption by *Chlorella vulgaris* at an initial Co^2+^ concentration of 41.3 mg/L: (**A**) cell growth expressed as OD_680_ values at different time points; (**B**) variation in cobalt concentration and corresponding removal efficiency as a function of contact time.

**Figure 7 ijms-26-10010-f007:**
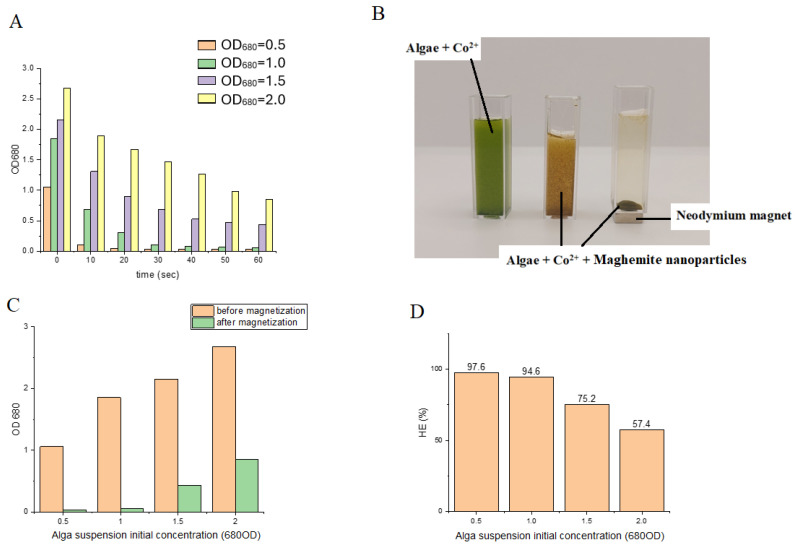
Magnetic separation of cobalt-adsorbed *Chlorella vulgaris*: (**A**) variation in UV–VIS spectra intensity at different times; (**B**) microscopic images of algal cells before and after magnetic separation; (**C**) changes in optical density of algal suspensions with different initial concentrations before and after magnetization; and (**D**) harvesting efficiency of algal suspensions with different initial concentrations.

**Figure 8 ijms-26-10010-f008:**
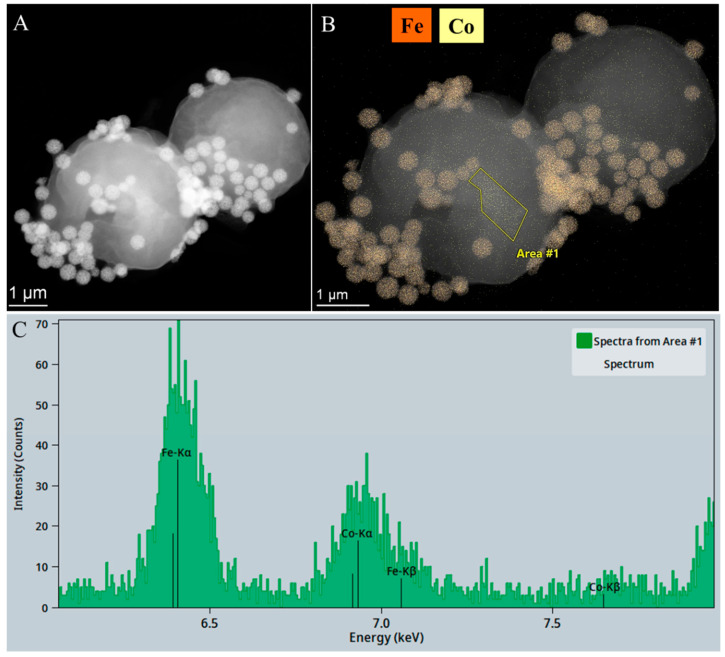
HAADF TEM pictures (**A**), element maps (**B**), and EDS spectrum (**C**) of the algae and the amine-functionalized maghemite nanoparticles after cobalt adsorption and magnetic separation.

## Data Availability

The datasets used and/or analyzed during the current study are available from the corresponding author on reasonable request.

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
