# Peer review of "Amine-Functionalized Maghemite Nanoflowers for Efficient Magnetic Removal of Heavy-Metal-Adsorbed Algae"

_ijms, 2025, doi:10.3390/ijms262010010_

Round 1

Reviewer 1 Report

Comments and Suggestions for Authors

 In this paper, the authors reported the synthesis of γ-Fe2O3 nanoparticles by solvothermal method, cobalt stock solution was add into the culture medium prior to inoculationt, then was inoculated with Chlorella vulgaris, the results showed that Co2+ removal efficiency achieving a 99±0.2 % within 48 hours. The article is logically coherent and reasonable data analysis. After reading the manuscript, I think it can be accepted for publication after revision.

1.The title of article should be shorter. For example, Algae based maghemite nanoflowers serve as highly effective adsorbents for removing heavy metals, or other.

2.Make sure all abbreviations are written out in full the first time used. This is particularly important in the abstract and in the conclusions, but work through the entire manuscript carefully from this perspective.

For instance, C. vulgaris(line 30) and Chlorella v. (line 282). Are they the same acronym?

3.Chlorella vulgaris are micron-sized, so whether the characterization and description in Figure 8 are correct?

4.Have you done a comparison experiment among maghemite nanoflowers, Chlorella vulgaris and Chlorella vulgaris based maghemite nanoflowers for cobalt ion removal? To exclude whether the addition of iron affects algal growth and to determine the adsorption subject.

5.Lines 24-26 of “The strong electrostatic interactions between functional groups (-NH2, -OH, -COOH) on maghemite particles and the negatively charged algal surface, enhanced by protonation at low pH, facilitated stable nanoparticle-biomass binding”. Is this conclusion applicable when pH value =6.5 is used in the paper?

Author Response

We are very grateful for the reviewer's careful work and comments and suggestions, to which we respond below:

In this paper, the authors reported the synthesis of γ-Fe2O3 nanoparticles by solvothermal method, cobalt stock solution was added into the culture medium prior to inoculation, then was inoculated with Chlorella vulgaris, the results showed that Co2+ removal efficiency achieving a 99±0.2 % within 48 hours. The article is logically coherent and reasonable data analysis. After reading the manuscript, I think it can be accepted for publication after revision.

  1. The title of article should be shorter. For example, Algae based maghemite nanoflowers serve as highly effective adsorbents for removing heavy metals, or other.

We thank for the suggestion regarding the title. However, we feel that the proposed version may lead to a misinterpretation, as it suggests that the nanoflowers are derived from algae and shifts the focus to direct heavy metal adsorption. Our study, in contrast, focuses on the magnetic removal of algae cells that have already bound heavy metals. To address the reviewer’s concern about length and clarity, we have revised the title to a shorter and more precise form, for example: ’Maghemite nanoflowers for efficient magnetic removal of heavy-metal-binding algae’. We believe this version more accurately reflects the scope and emphasis of the manuscript.

  1. Make sure all abbreviations are written out in full the first time used. This is particularly important in the abstract and in the conclusions, but work through the entire manuscript carefully from this perspective.

We thank for this useful suggestion. We have carefully checked the manuscript to make sure that all abbreviations are spelled out in full when they first appear and are used in the same way throughout the text.

  1. For instance, C. vulgaris(line 30) and Chlorella v. (line 282). Are they the same acronym?

We thank for pointing out this inconsistency. Indeed, both abbreviations referred to Chlorella vulgaris. We have corrected the manuscript to use a consistent notation throughout.

  1. Chlorella vulgaris are micron-sized, so whether the characterization and description in Figure 8 are correct?

We thank for this observation. Chlorella vulgaris cells are indeed in the micron size range, which is consistent with the scale bar shown in Figure 8. In this image, the larger spherical structures correspond to the algal cells, while the smaller bright dots visible on their surface represent the attached maghemite nanoparticles. The figure was intended to illustrate the successful binding of nanoparticles to the algal cell surface rather than to provide a high-resolution morphological characterization of the algal cells themselves. To avoid misunderstanding, we have revised the corresponding text in the manuscript to clarify this point.

  1. Have you done a comparison experiment among maghemite nanoflowers, Chlorella vulgaris and Chlorella vulgaris based maghemite nanoflowers for cobalt ion removal? To exclude whether the addition of iron affects algal growth and to determine the adsorption subject.

We appreciate the insightful comment. In this study, we did not compare maghemite nanoflowers, Chlorella vulgaris, and Chlorella vulgaris-based maghemite nanoflowers to see which one was best at removing cobalt ions. The main aim of our work was to show that it is possible to quickly and efficiently separate algae cells that have already been linked with heavy metals using magnets. We did not aim to investigate how well this worked or what the effects of adding iron were on the growth of algae. We agree that these kinds of studies would provide extra useful information, and we plan to include this in future research.

  1. Lines 24-26 of “The strong electrostatic interactions between functional groups (-NH2, -OH, -COOH) on maghemite particles and the negatively charged algal surface, enhanced by protonation at low pH, facilitated stable nanoparticle-biomass binding”. Is this conclusion applicable when pH value =6.5 is used in the paper?

We thank for this insightful question. Yes, this conclusion also applies at pH 6.5. Although the measured zeta potential of the amine-functionalised maghemite suspension at this pH was slightly negative (ζ =–2.4 to -10 mV), the primary amine groups (pKₐ ~9–10) remained largely protonated. This provided localised positive charges that interacted with the negatively charged algal cell wall (ζ = –20.1±0.4 mV). In addition to electrostatic interactions, stable nanoparticle–biomass binding is also facilitated by hydrogen bonding between the surface functional groups of the nanoparticles and the components of the algal cell wall. We have revised the manuscript to clarify this accordingly.

Reviewer 2 Report

Comments and Suggestions for Authors

In the submitted paper by Vanyorek et al., it is demonstrated how biomass (algae) can be used for the effective removal of toxic metal ions. The idea is well presented and has potential for industrial implementation. However, this reviewer has observed several shortcomings in the experimental studies, which need to be addressed and improved before the paper can be considered for publication.

  • The authors should attempt to remove Co²⁺ ions across a wide pH range of water. This is one of the major shortcomings of the paper. Although the acceptable pH values for wastewater are mentioned, in practice, pH often fluctuates, which is a very important issue. The authors need to perform removal experiments across the entire pH range and compare the results with those already obtained.
  • Additionally, it is unclear why the authors did not test other metal ions, since in real wastewater samples there is often competition between ions for binding sites. This aspect should be addressed by including experiments with other relevant metal ions, such as nickel and manganese.
  • Also, while reading the paper, it was not immediately clear that the authors worked with modified Fe nanoparticles. This information only becomes apparent at the end of the manuscript, in the Materials and Methods section (Section 3.2: Synthesis of the amine-functionalized maghemite nanoflowers). This important detail should be clearly stated earlier and more consistently throughout the manuscript.
  • Additionally, at the beginning of the paper, the authors state that the algal cell wall possesses functional groups important for binding to nanoparticles through electrostatic interactions (Page 2, line 66: “Specifically, the algal cell wall incorporates carboxyl (-COOH), hydroxyl (-OH), amino (-NH₂), carbonyl (-C=O), ester (-CO–O–), sulfhydryl (-SH), sulphate (-SO₄²⁻), and phosphate (-PO₄³⁻) groups”). Later, on Page 5, line 157, the authors write: “The presence of -NH₂, -OH and -COOH functional groups on the surface of maghemite particles is key important, because these groups contribute to the adsorption processes between the magnetic nanoparticles and algae cells.” This raises confusion: are these functional groups located on the Fe nanoparticles, on the algal cell surface, or on both? And more importantly, which of them are primarily responsible for the binding interaction? Please clarify this point, as it is essential for understanding the adsorption mechanism.

Additional remarks:

Page 5, line 161, 52cm-1???

Figure 6A needs to be better explained by adding the exact values for both the control and the sample with Co²⁺ directly in the graph. Additionally, the concentration of Co²⁺ used in the experiment should be clearly stated in the caption of Figure 6.

Author Response

We are very grateful for the reviewer's careful work and comments and suggestions, to which we respond please find attached. 

Reviewer 3 Report

Comments and Suggestions for Authors

The article submitted for review is devoted to the study of the sorption properties of freshwater algae Chlorella vulgaris and γ-fe₂o₃ for the removal of cobalt ions. The topic of the article is relevant and corresponds to the profile of the journal. However, the authors need to make some clarifications.

  1. The data in Figure 5 do not allow us to identify γ-fe₂o₃, the nature of the dependence is more suitable for α-fe₂o₃. The authors need to obtain the Mossbauer spectra and compare them with the spectra for nanocrystalline hematite α-Fe2O3, maghemite γ-Fe2O3 and magnetite Fe₃o₄.
  2. Chlorella vulgaris can form insoluble complexes with heavy metals, including cadmium. It is necessary to explain how the regeneration or disposal of spent biomaterial will be carried out.

Author Response

We are very grateful for the reviewer's careful work and comments and suggestions, to which we respond below: 

The article submitted for review is devoted to the study of the sorption properties of freshwater algae Chlorella vulgaris and γ-fe₂o₃ for the removal of cobalt ions. The topic of the article is relevant and corresponds to the profile of the journal. However, the authors need to make some clarifications.

The data in Figure 5 do not allow us to identify γ-fe₂o₃, the nature of the dependence is more suitable for α-fe₂o₃. The authors need to obtain the Mossbauer spectra and compare them with the spectra for nanocrystalline hematite α-Fe2O3, maghemite γ-Fe2O3 and magnetite Fe₃o₄.

Thank you for the suggestion, the Mössbauer spectroscopy is indeed a very effective tool for examining magnetic materials, but we do not have the facilities to perform such measurements. However, we reliably identified the phase composition of the magnetic sample using Rietveld refinement and X-ray diffraction. The amount of maghemite (γ- Fe2O3) was 91.8 wt%, alongside 8.2 wt% magnetite (Fe3O4). Hematite (α-Fe2O3) was not found in the sample. Maghemite was also verified by another test method, selected area electron diffraction measurement (SAED), and by evaluating the diffraction rings, we determined the lattice spacing (d spacing), which shows good agreement with the values found in the X-ray database (PDF No. 39-1346). The high saturation magnetization of the sample also suggests that the phases present can only be well magnetizable iron oxides, namely maghemite and magnetite.

Chlorella vulgaris can form insoluble complexes with heavy metals, including cadmium. It is necessary to explain how the regeneration or disposal of spent biomaterial will be carried out.

We thank for this valuable suggestion. As recommended, we have revised the manuscript to include a discussion on possible strategies for the regeneration and/or safe disposal of heavy-metal-loaded algal biomass. Specifically, we now highlight approaches such as desorption of metals for biomass reuse, as well as immobilization or thermal treatment for safe disposal.

Round 2

Reviewer 2 Report

Comments and Suggestions for Authors

The manuscript can be accepted for publication.

Reviewer 3 Report

Comments and Suggestions for Authors

It is a pity that the authors do not have the opportunity to study the material using Mossbauer spectroscopy. Nevertheless, the results obtained earlier allow us to hope for a correct identification of the phase composition.